# Liquefaction of water on the surface of anisotropic two-dimensional atomic layered black phosphorus

Jinlai Zhao[1,2,3], Jiajie Zhu[3], Rui Cao[1,2], Huide Wang[2], Zhinan Guo[2], David K. Sang[2], Jiaoning Tang[3], Dianyuan Fan[2], Jianqing Li[1] & Han Zhang [2]

The growth and wetting of water on two-dimensional(2D) materials are important to understand the development of 2D material based electronic, optoelectronic, and nanomechanical devices. Here, we visualize the liquefaction processes of water on the surface of graphene, $MoS_2$ and black phosphorus (BP) via optical microscopy. We show that the shape of the water droplets forming on the surface of BP, which is anisotropic, is elliptical. In contrast, droplets are rounded when they form on the surface of graphene or $MoS_2$, which do not possess orthometric anisotropy. Molecular simulations show that the anisotropic liquefaction process of water on the surface of BP is attributed to the different binding energies of $H_2O$ molecules on BP along the armchair and zigzag directions. The results not only reveal the anisotropic nature of water liquefaction on the BP surface but also provide a way for fast and nondestructive determination of the crystalline orientation of BP.

[1] Faculty of Information Technology, Macau University of Science and Technology, Avenida Wai Long, Taipa, Macau 999078, PR China. [2] Shenzhen Engineering Laboratory of Phosphorene and Optoelectronics, International Collaborative Laboratory of 2D Materials for Optoelectronics Science and Technology, Engineering Technology Research Center for 2D Material Information Function Devices and Systems of Guangdong Province, Institute of Microscale Optoelectronics (IMO), Shenzhen University, Shenzhen 518060, PR China. [3] College of Materials Science and Engineering, Shenzhen Key Laboratory of Polymer Science and Technology, Guangdong Research Center for Interfacial Engineering of Functional Materials, Shenzhen 518060, PR China. Correspondence and requests for materials should be addressed to Z.G. (email: guozhinan@szu.edu.cn) or to H.Z. (email: hzhang@szu.edu.cn)

Black phosphorus (BP), one of the most stable allotropes of phosphorus[1,2], has recently (since 2014) emerged as an important two-dimensional (2D) material[3,4]. BP consists of corrugated planes of phosphorus atoms with extremely strong intralayer bonding and weak interlayer interactions[5]. Remarkably, as a metal-free layered semiconductor, the layer-dependent direct band gap of BP is tunable from 2 eV for a monolayer to 0.3 eV for the bulk material[5,6]. Furthermore, layered BP possesses extremely high carrier mobility (~200–1000 cm$^2$ V$^{-1}$ s$^{-1}$) and a moderate on/off ratio (~10$^4$–10$^5$)[7,8], holding extensive applications in electronic and optoelectronic devices[9], such as field-effect transistors (FETs)[3,10,11], photodetectors[12], and photothermal agents[13–15]. In contrast to graphene and MoS$_2$, BP is a 2D material that possesses in-plane anisotropy due to its special intrinsic crystallographic structure caused by its differing bond angles and bond lengths along its orthogonal in-plane directions. Many anisotropic behaviors of BP, including the optical, vibrational, electronic, thermal, and mechanical aspects[1,2,9,16–22], have been studied. Such anisotropic properties of BP play a vital role in designing polyfunctional and controllable 2D innovative electronic, optoelectronic, and nanomechanical devices, which are impossible for other isotropic 2D layered materials[23].

Many experiments and theoretical calculations have been carried out to study the anisotropic properties, and methods have been developed to identify the specific armchair (AC) and zigzag (ZZ) crystalline directions of BP. The electrical and thermal conductance values of layered BP exhibit strong spatial anisotropies. Their respective preferred directions of conductance are mutually orthogonal, leading to an anisotropic electrical[3,17,24] and thermoelectric[2,9,25] figure of merit, which is larger along the AC direction. The intrinsic anisotropic light–matter interactions in BP, including the electron–photon interactions, make anisotropic optical absorption and scattering spectroscopy a reliable and simple in situ way to identify the crystalline orientation of BP[21]. By comparing the ratio of the intensity of the Raman peaks, angle-resolved polarized Raman spectroscopy has become an accurate method to identify the specific ZZ and AC crystalline directions of BP[19,24]. In addition to optical methods, mechanical methods can also be applied to distinguish the direction of BP, because the values of the Young's modulus, breaking stress, and elastic modulus of BP are all higher in the ZZ direction than in the AC direction[22,26]. However, these methods either rely on expensive equipment, sophisticated data collection and analysis, complex experimental design, or cause some damage to the structure of BP. In this case, a simpler method should be developed for fast in situ identification of the orientation of BP, which does not rely on expensive and complicated equipment.

The growth and motion of atoms, molecules, and clusters on a crystal surface is always an important research topic in the field of surfaces and interfaces. After 2D materials gained popularity since the discovery of graphene, studies on the behaviors of water wetting and diffusion on or in atomic structured layers became possible. Recently, a series of theoretical simulations regarding the behaviors of water on graphene[27–29], boron nitride[30–32], and MoS$_2$[33–35] were published, and a mechanism for the surface diffusion of water on these materials was determined. However, experimental results on this topic are rarely reported, especially for atomic layered materials with anisotropic properties, even though the anisotropic wetting characteristics of water droplets on black phosphorene has already been theoretically predicted[36,37].

Herein, the liquefaction process of water on the surface of BP, which is a typical atomic layer with anisotropic properties, has been first visualized by a microscopy. It has been observed that the shape of the water droplets forming on the surface of BP is mainly elliptic rather than rounded, as they form on the surfaces of materials not possessing orthometric anisotropy, including graphene and MoS$_2$. The anisotropic liquefaction process of water on the surface of BP is attributed to the differing binding energies of H$_2$O molecules on BP along the AC and ZZ directions. The results not only disclose the anisotropic nature of water liquefaction on the BP surface but also provide a way for fast and nondestructive determination of the crystalline orientation of BP, which is important for future research on anisotropic BP-based electronic, optoelectronic, and nanomechanical devices.

## Results

**Liquefaction of water on BP, graphene, and MoS$_2$.** To investigate the liquefaction process of water on the surface of BP, a vapor generator and an optical microscope were used[38], as shown in Supplementary Fig. 1. When the gas phase water coming out from the generator reaches the surface of the Si/SiO$_2$ wafer carrying BP flakes, a phase change will happen there (gas phase water changes into the liquid phase). Small droplets will grow large and aggregate as time goes on. When the size of the droplets is large enough, they can be observed by a microscope. Both images and videos of the liquefaction progress of the vapor on the surface of BP flakes were recorded by an optical microscope with

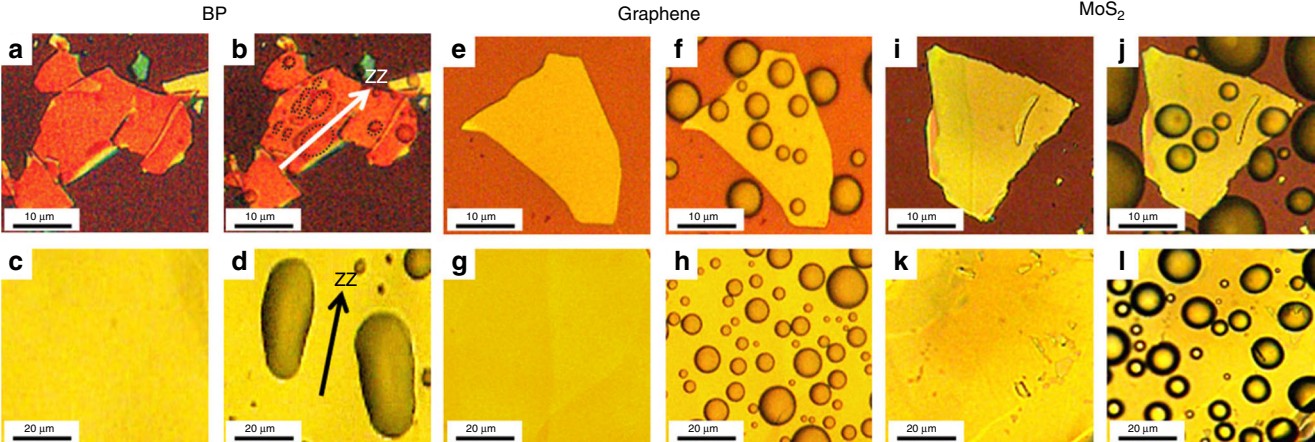

**Fig. 1** Optical microscope images of BP/graphene/MoS$_2$ without and with water droplets. **a**, **e**, **i** Few-layer BP/graphite/MoS$_2$ without water droplets, **b**, **f**, **j** few-layer BP/graphite/MoS$_2$ with water droplets, **c**, **g**, **k** thick BP/graphite/MoS$_2$ without water droplets, and **d**, **h**, **l** thick BP/graphite/MoS$_2$ with water droplets

2 K magnification, as shown in Fig. 1 and Supplementary Movie 1.

Supplementary Movie 1 shows that water vapors first find the condensation nucleus on the BP surface. Then, the condensation progress happens around the nucleus, which leads to very tiny water droplets forming, and the tiny water droplets will become bigger and bigger as time goes on. Statistics, as shown in Supplementary Fig. 2, have been conducted at 12, 13, 14, and 15 s in Supplementary Movie 1 to demonstrate the shape evolution of water droplets during the growth process. It demonstrates that the water droplets on the surface of BP layer are either elliptical or spherical, but ellipsoids are dominant. In addition, as is shown in Supplementary Fig. 2e, the ratios of long axis and short axis for the elliptical water droplets range from 1.2 to 3.0, mainly 1.8 as time goes on. Importantly, all long axes of the water droplets point in almost the same direction, as shown in Fig. 1b, d.

However, such anisotropic condensation progress of water vapors can only be observed on the surface of anisotropic BP. On an isotropic surface, such as graphite and $MoS_2$, the anisotropic shape of water droplets cannot be observed, as shown in Fig. 1. In contrast, the water droplets on graphene and $MoS_2$ nanosheets are totally round. In addition, Supplementary Movies 2 and 3 show that throughout the whole condensation progress, the water droplets on the surface of graphene and $MoS_2$ are all round rather than elliptical. In summary, the elliptic water droplets can only be observed on the anisotropic surface. In addition, the phenomenon of forming elliptic water droplets on the anisotropic atomic stacked structure surface has nothing to do with the atomic layer thickness of the sample. Figure 1c, d shows that the elliptic water droplets can also be formed on the thick BP surface, but the water droplets on thick graphite and $MoS_2$ sheets remain round (see Fig. 1h,1l).

**Direction of elliptic water droplets on BP**. To determine the relationship between the distribution directions of the water droplets on the surface of BP sheets and the anisotropic atomic structure of BP, a polarized Raman experiment has been conducted according to the method reported previously[9,19]. The peak positions of all three typical vibrational modes ($A_g^1$, $B_{2g}$, and $A_g^2$) did not change as the excitation laser polarization angle varied from 0° to 180°, as shown Fig. 2a. However, all the intensities of the $A_g^1$, $B_{2g}$, and $A_g^2$ modes vary periodically with the incident laser polarization changing from 0° to 180°, which shows obvious angle-dependent anisotropy for BP sheets and is highly consistent with the previous reports[19,20,24]. It has been demonstrated that the relatively larger local maximum intensities of $A_g^1$ mode peaks corresponds to the AC directions of BP, while the relatively smaller local maximum associates with the ZZ directions[19]. In this case, a polar diagram based on the intensities of the $A_g^1$ mode could be obtained by tuning the angle of the polarization of the incident laser. Based on the polar diagram shown in Fig. 2b, the AC and ZZ directions could be determined. By using this polarized Raman method, the long axis of the water droplets has been proven to be along the ZZ direction. To further confirm this consistency, the polarized Raman measurements and the liquefaction of water experiments were repeated ten times. This finding shows that the difference of the direction of the elliptic water droplets and ZZ direction of BP is <10°. In this case, compared to the angle-resolved polarized Raman spectroscopy, which relies on expensive equipment and complex data collection and analysis, observation of the liquefaction of water on BP could be a much cheaper, faster, and more convenient method to identify the AC and ZZ directions of BP sheets.

**Mechanism and simulations for anisotropic wetting on BP**. One of the physical factors of anisotropic wetting is ascribed to the liquid contact line encountering physical discontinuities such as the sharp solid edges. For example, nanometer- and micrometer-scale surface structures (groves, parallel lines, pillars, wrinkles) can lead to macroscopic, visible changes in anisotropic wetting behaviors[39]. Herein, the anisotropic wetting phenomena of BP could be explained by the anisotropic structure of phosphorene. It is well known that the arrangement of BP atoms makes the BP layer a puckered and anisotropic structure, as the schematic diagram of BP shows in Fig. 3, which graphene and $MoS_2$ do not exhibit. Because the water droplets are suspended on the puckered structure and do not directly contact the bottom of the BP atomic substrate, based on the lattice spacing of BP and the size of each $H_2O$ molecule, the Cassie model has been used to explain the anisotropic wetting behavior of water[40]. The puckered structure of BP exhibits a larger pinning effect of water drops along the AC direction than along the ZZ direction for water liquefaction. The larger pinning effect along the AC direction results in a larger restriction for both wetting and water diffusion. Therefore, the energy barrier for diffusion of water along the AC direction is much larger than that along the ZZ direction[41]. It is obvious that the water droplets move faster along the ZZ direction than along the AC direction when the water vapors condense on the surface of BP, which leads to the water droplets forming ellipses on the BP surface. Importantly, the directions of the long axes of the elliptical water droplets are parallel to the ZZ direction of BP, as shown in Fig. 3b.

From an absorption energy perspective regarding the anisotropic wetting on layered BP, ab initio electronic structure

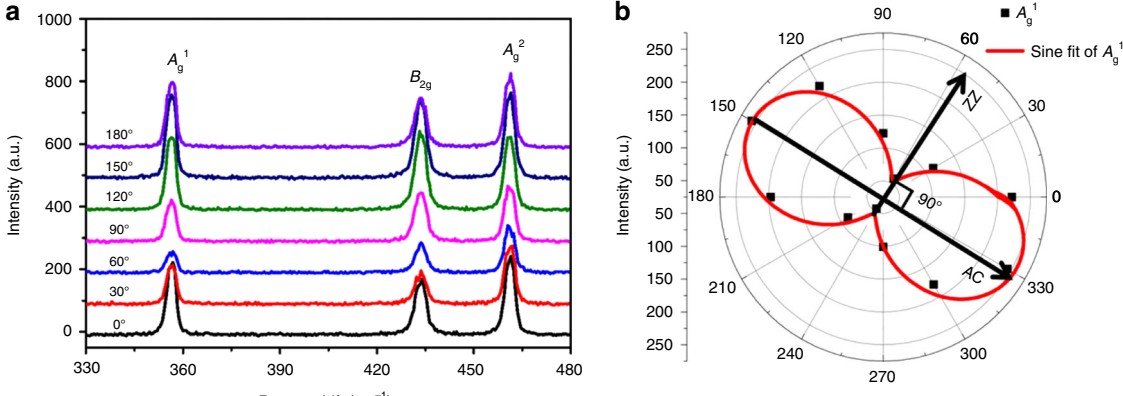

**Fig. 2** Polarization-resolved Raman scattering spectrum. **a** Raman curves with different incident angles from 0° to 180° with a 632 nm laser. **b** Polar plots of the fitted peak intensities of the $A_g^1$ modes as a function of sample rotation angle under parallel cross-polarization configurations

calculations and molecular dynamics simulations were conducted. The calculation results show that the binding energy of $H_2O$ molecules on phosphorene is 0.17 eV, reflecting a weak interaction, which leads to a large vertical distance of 3 Å between

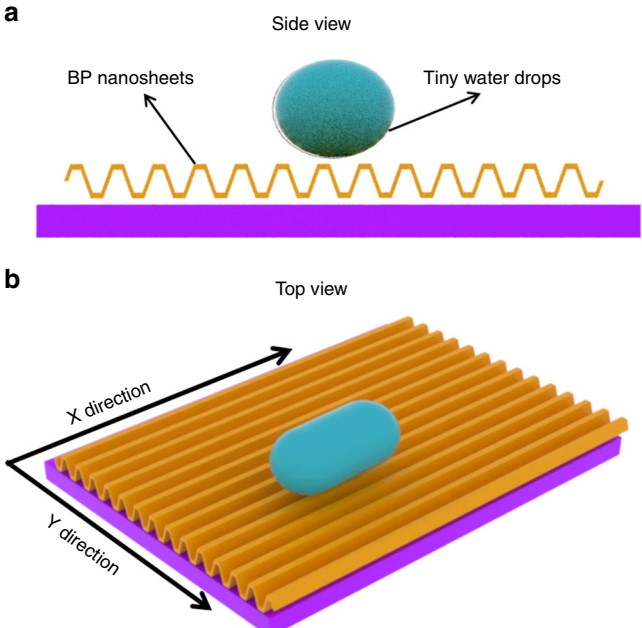

**Fig. 3** The schematic diagram of water droplet on the surface of BP. **a** Side view and **b** top view. The *X* direction is in the zigzag structure and the *Y* direction is in the armchair structure

the molecule and the sheet, which is in agreement with the previous simulation results[37]. The $H_2O$ molecule almost retains the original structure. In addition, six configurations are considered for $H_2O$ molecules on phosphorene, namely, along the AC, ZZ, and diagonal directions, as shown in Fig. 4. Configuration D, showing a large distortion, turns out to be the ground state, with an energy difference of 0.098–0.275 eV molecule$^{-1}$ compared to other configurations. Thus, $H_2O$ molecules are predicted to align along the ZZ direction on phosphorene. The result is in accordance with the previous calculation results[36,37], proving that the energy barrier for diffusion of the water droplets along the AC direction is larger than that along the ZZ direction for BP layers, which leads to anisotropic diffusion of water droplets. The anisotropic diffusion of water droplets results in anisotropic shape of water droplets, which normally are elliptical on the surface of BP layer just like the optical microscope images showing in the experiment. Moreover, the same simulations of $H_2O$ molecules on graphene and $MoS_2$ have been carried out as well. The results demonstrate that both graphene and $MoS_2$ possess a ground state among the six configurations for $H_2O$ adsorption, with an energy difference of 0.008–0.063 eV molecule$^{-1}$ for graphene and 0.003–0.051 eV molecule$^{-1}$ for $MoS_2$, as shown in Supplementary Figs. 4 and 5. However, compared with BP, the energy differences of graphene and $MoS_2$ are much smaller, which leads to inconspicuous anisotropy, and is consistent with the earlier experiment results.

**Nondestructive liquefaction of water on BP.** As we know that degradation occurs when BP layers are exposed to oxygen and water in air[5,42–45], so all the wetting experiments were conducted in a glove box with the protection of an inert gas (Ar), as shown in Supplementary Fig. 1. Meanwhile, the BP layers can retain

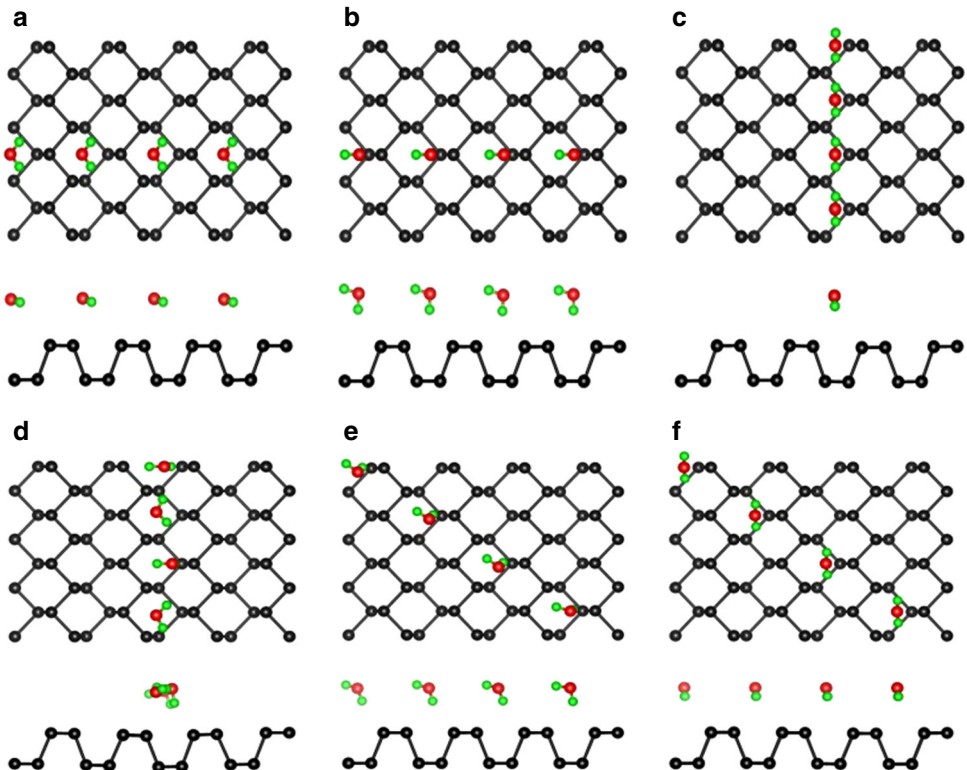

**Fig. 4** Six representative high-symmetry configurations for $H_2O$ molecules on phosphorene. **a, b** Along armchair direction, **c, d** along zigzag direction, and **e, f** along diagonal direction, and the total energy difference is **a** 0.104 eV molecule$^{-1}$, **b** 0.098 eV molecule$^{-1}$, **c** 0.275 eV molecule$^{-1}$, **d** 0 eV molecule$^{-1}$, **e** 0.123 eV molecule$^{-1}$, and **f** 0.150 eV molecule$^{-1}$, respectively

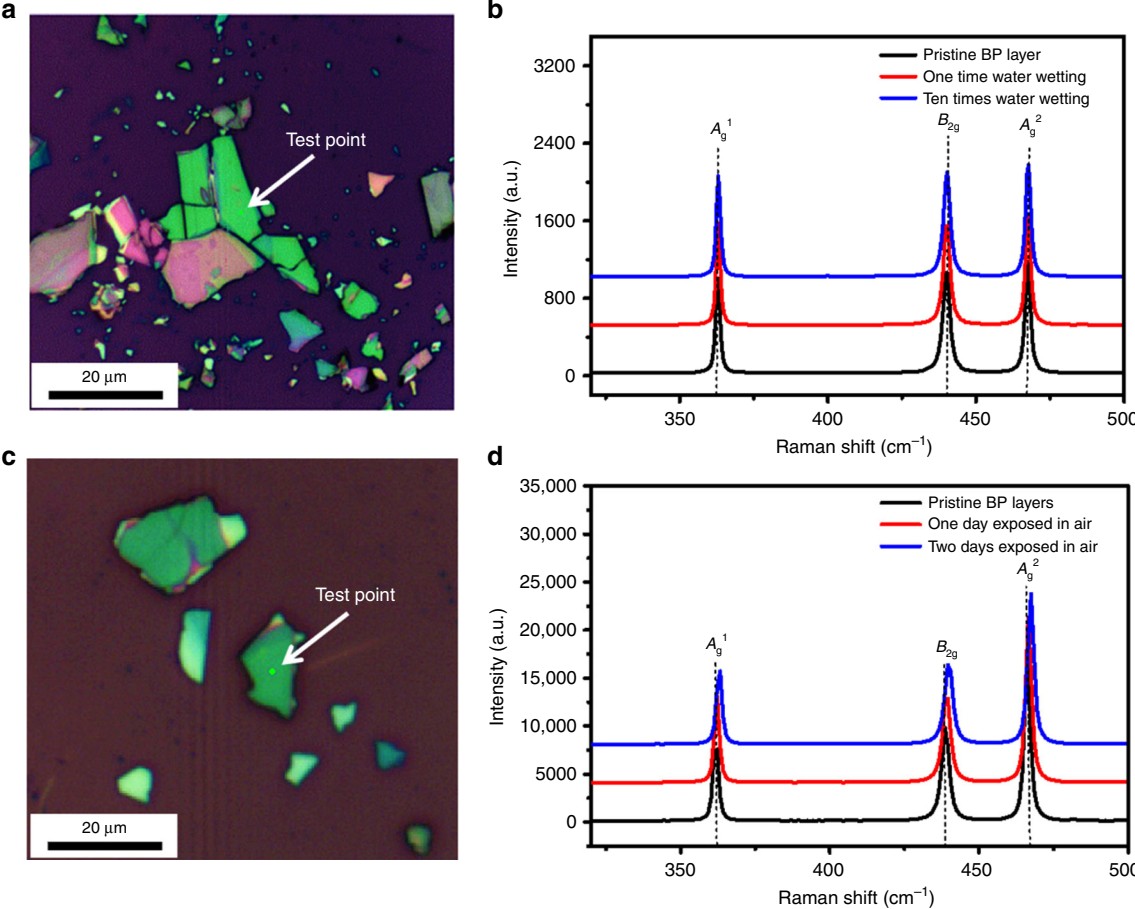

**Fig. 5** In situ Raman spectroscopy test of few-layer BP. **a**, **c** Optical microscope image of BP layers, **b** Raman curves of BP nanosheets: pristine (black), after one water wetting experiment (red), and after ten water wetting experiments (blue), and **d** Raman curves for pristine BP layers and BP layers 1 and 2 days after exposure to air

their intrinsic properties commendably during the wetting test. As shown in the AFM images of Supplementary Fig. 3, the surfaces of the BP layers almost remain the same without any bubbles after ten wetting experiments. The zoomed AFM image in the yellow dashed line frame (Supplementary Fig. 3c) shows that the Ra can reach 0.22 nm, which is another proof that there was no oxidation reaction on the BP surface. To further demonstrate the nondestructive performance of BP layers in the course of the wetting test, Raman spectroscopy tests in situ for pristine BP layer and for samples after one water wetting experiment and ten water wetting experiments were carried out, as shown in Fig. 5c. All the Raman peak positions and peak intensities of the $A_g^1$, $B_{2g}$, and $A_g^2$ peaks at the same test point are almost not changed, which clearly indicates the stable and non-destructive property of BP layers after ten water wetting experiments. In contrast, the Raman curves of pristine BP layers and BP layers 1 and 2 days after exposure to air, as shown in Fig. 5d, clearly demonstrate that all three peaks at ~363, 440, and 466 cm$^{-1}$, corresponding to the $A_g^1$, $B_{2g}$, and $A_g^2$ Raman modes of BP, respectively, move a little to the right, and all the peak intensities decrease. Importantly, the AFM images of the surface of BP layers 1 and 2 days after exposure to air have many bubbles, as shown in Supplementary Fig. 3d, e. Both the Raman and AFM imaging data obviously show the degradation of BP layers[43] in air for a period of time. Furthermore, a few-layer BP-based FET device (~8.34 nm) was fabricated; as shown in the inset of Supplementary Fig. 6b, the transport performance has been tested before and after water liquefaction experiment[46]. The result shows that the $I_d$–$V_g$ curves almost remain the same before (black line) and after ten wetting experiments (red line), which also demonstrates that there was no structural collapse existing on the BP after ten wetting experiments. In summary, this certainly confirms that the method to identify the anisotropy of BP sheets by using the wetting test is completely nondestructive. It only takes about 10 s to identify the BP atomic orientation without destroying the intrinsic atomic structure of BP, which is very important for the application of anisotropic BP-based electronic, optoelectronic, and nanomechanical devices.

## Discussion

In summary, the liquefaction process of water vapors on the surface of 2D materials was observed via optical microscope. The shapes of the water droplets on the surface of layered BP, possessing anisotropic atomic arrangement, are totally elliptical through all the condensing progress. On the contrary, the shape of the water droplets is apparently round on the isotropic atomic arrangement surface, such as graphene and layered $MoS_2$. One of the physical factors of anisotropic wetting is ascribed to the pinning effect of the water droplets flowing on the BP surface possesses larger energy barrier along the AC direction than along the ZZ direction. From the perspective of absorption energy, the arrangement of water along the ZZ direction of BP needs the smallest binding energy. The method to study the liquefaction process of water vapors on BP surface can be used for fast identification of the orientation of BP structure. Without any

large-scale instruments and complicated data analyses, the accurate orientation information of BP can be obtained within 10 s. Moreover, this method can be promoted to some other 2D materials possessing both in-plane and out-of-plane anisotropic properties. The study paves a way for future research on anisotropic 2D material-based electronic, optoelectronic, and nanomechanical devices.

## Methods

**Sample preparation**. All 2D material flakes (BP, graphene, and MoS$_2$) were produced from their bulk crystals by a scotch tape-based mechanical exfoliation method and then transferred onto a Si/SiO$_2$ (300 nm) wafer by a polydimethylsiloxane film as the medium. During the experiments, all 2D material flakes were characterized immediately after they had been prepared to ensure that all obtained properties are intrinsic, without any structure degradation and contaminations. Additionally, all the few-layered materials studied here were in 8–10 nm thickness, because 2D layered materials in such thickness had been most extensively studied and possess relatively better performance for devices research than monolayer ones, especially for BP layers[3,47–50].

**Water droplets forming on the nanosheets**. A vapor generator from the Alibaba company had been applied for vapor generation. After the SiO$_2$/Si substrates with different kinds of 2D material flakes were fixed on a flat board, the outlet of the generator facing the sample was turned on. The output vapor liquefied on the surface of the 2D material flakes, and microscale droplets formed there. By using an optical microscope, the size and shape of the droplets could be clearly observed. The whole growing process of each droplet was observed and recorded. To prevent the reaction between the BP samples and the oxygen in the air, which would influence the anisotropic property of BP, the whole process was carried out in a glove box under the protection of an argon atmosphere.

**Polarization-dependent Raman characterization**. Polarized Raman scattering spectroscopy was performed with a homemade Raman system. A polarization analyzer was put between an edge filter and the detector to achieve parallel polarization, while the cross-polarization configuration was obtained by putting a half-wave plate in the incident laser path. Then, polarized Raman spectra with different angles, $\alpha$, were achieved by rotating the half-wave plate with $\alpha/2$ from the parallel polarization configuration[19].

**Computational method**. The total energy calculations were performed in the framework of density functional theory using the projector-augmented wave method as implemented in the Vienna Ab initio Simulation Package[51]. The generalized gradient approximation of Perdew, Burke, and Ernzerhof was selected for the exchange-correlation potential[52]. The long-term van der Waals interaction was taken into account by the DFT-D3 approach[53]. The cut-off energy for plane-wave basis sets was set to 500 eV. A $4 \times 4 \times 1$ supercell was used to model H$_2$O molecule adsorption on phosphorene. A $2 \times 1 \times 1$ $k$-mesh was used for the Brillouin zone integrations. The structures were relaxed until the residual forces on the atoms declined to <0.01 eV A$^{-1}$.

## Data availability

All data that support the findings of this study are available from the corresponding author upon reasonable request.

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

## Acknowledgements

We would like to thank Dr. Yani Chen and Prof. Mingyuan Huang from the Department of Physics in the Southern University of Science and Technology (SUST) for their efforts on the polarized Raman measurements. We authors acknowledge financial support from the National Natural Science Fund (Grant Nos. 61605131, 61435010, and 51778369), Guangdong Science Foundation for Distinguished Young Scholars (2018B030306038), Science and Technology Innovation Commission of Shenzhen (Grant Nos. JCYJ20180507182047316, KQJSCX2018032809550179, KQTD2015032416270385, JCYJ20150625103619275, and ZDSYS201707271014468), Educational Commission of Guangdong Province (2016KCXTD006), and the Science and Technology Development Fund (Grant No. 007/2017/A1 and 132/2017/A3), Macao SAR, China.

## Author contributions

J.L.Z., Z.G., and H.Z. conceptualized this work; J.L.Z. developed the methodology; J.L.Z. carried out the water wetting test, Raman measurements, and AFM investigation; J.J.Z. made the simulation calculation; R.C., H.W., and D.K.S. help to process the data; J.L.Z. wrote the original draft; all authors reviewed and edited the manuscript; Z.G. and J.T. acquired funds; H.Z., J.L., and D.F. supervised this work.

## Additional information

**Competing interests:** The authors declare no competing interests.

**Peer Review Information:** *Nature Communications* thanks Gang Zhang and other anonymous reviewer(s) for their contribution to the peer review of this work. Peer reviewer reports are available.

