## [Peer Review File · Nature Communications]

Reviewers' comments:

Reviewer #1 (Remarks to the Author):

This paper presented a systemic study on the liquefaction process of water on black phosphorus surface based on experimental and simulation methods, which was also compared with that on graphene and MoS₂ surfaces. The experimental observations showed that the water droplets formed elliptical shapes on black phosphorus, while they were rounded on graphene and MoS₂. Furthermore, molecular simulation results indicated that the anisotropic liquefaction process of water on black phosphorus is attributed to the different binding energies of water molecules on black phosphorus along the armchair direction and zigzag direction. This manuscript is suitable for publishing on Nature Communications. However, there are still some major modifications needed.

1. Figure 1 shows some elliptical shapes of water on black phosphorus. I suggest adding more detailed analysis on the shape evolution and size effect, such as the quantitative data on the shape evolution during the growth process of a water droplet. Even though they are always elliptic during growth, the difference between long axes and short axes of elliptical shape should be different, which depends on the size of water droplet.
2. Figure 4 indicates that water molecules prefer to align along the zigzag direction than armchair direction since there is an energy difference of 0.1-0.3 eV/molecule between them. It is more convincing to perform some simulations of water on graphene and compare the results of graphene with phosphorene.
3. Line 90 page 5 mentioned that the anisotropic wetting of water on phosphorene has already been theoretically predicted. However, no citation on the theoretical study of anisotropic wetting of phosphorene is given. It is better to cite some simulation studies on the anisotropic wetting of phosphorene (J. Phys. Chem. C 2018, 122, 4622–4627; Sci. Rep. 2016, 6, 38327) and compare your experimental observations with them, such as the similarities and differences on the shape and energy of water.

Reviewer #2 (Remarks to the Author):

In this manuscript, the authors observed the liquefaction of water on the surfaces of various 2D materials, including black phosphorus (BP), MoS₂ and graphite. Elliptical water droplets were observed on the surface of BP crystal, which was attributed to the anisotropic atomic layer of BP. While rounded droplets were formed on the surfaces of MoS₂ and graphite crystals. This work

shows a new approach to determine the crystalline orientation of BP without expensive instrument. The results are novel and should attract interest from many researchers in the fields of 2D materials. However, there are several points that need to be clarified before this work is ready for publication.

1. In the experiment, water vapor was generated and transported to the surface of 2D materials. It is well known that oxygen is dissolved in water at ambient conditions. Thus it is possible that photo-induced oxidation reaction could occur on BP surface in the presence of oxygen and water. The authors need to clarify the influence of oxidation on liquefaction in this process. Meanwhile, AFM images are necessary to confirm it before and after liquefaction. I cannot find this evidence in current form, which weakens the quality of this manuscript.

2. Previously, researchers have found that rounded and irregularly shaped water droplets were formed on BP surface during microdroplet condensation experiment (2D Mater. 1 (2014) 025001; 2D Mater. 2 (2015) 011002). These data are inconsistent with this work. The authors should justify and clarify it.

3. In line 179, the authors claimed that the BP surface had no bubble remained after 10 times wetting experiments. However, no detailed or zoomed AFM image was provided to confirm it. In addition, there is a new small flake appeared between the two flakes, located at the left of white line. Possibly it comes from a contaminated AFM tip. In this case, the data should be replaced to avoid misunderstanding.

4. In figure 5, only Raman data are not enough to demonstrate the nondestructive performance of liquefaction for BP surface. The authors should provide more characterization data to support it, such as AFM image.

5. In line 190, the authors claimed many bubbles were left on BP surface after exposure to air for 2 days. However, these structures were attributed to water droplets in the literatures (2014) 025001; 2D Mater. 2 (2015) 011002). How can the authors prove they are bubbles?

6. Although the idea is interesting, I think the work has weakness in the characterization for surface investigation. In my opinion, AFM and other instruments should be necessary to monitor the evolution before and after liquefaction to provide solid evidence. Only optical microscope and Raman spectrometer are insufficient. The authors almost reveal the final result but lots of works are remained to be conducted.

7. Correct the format of reference.

Reviewer #3 (Remarks to the Author):

This work reports that the shape of the water droplets formed on the surface of black phosphorus is elliptic in contrast to rounded, as they form on the surfaces of other 2D materials, such as graphene

and MoS2 which is orthometric. The authors explain that this difference could allow for a fast characterization of the lattice orientation of BP.

In Fig 1 the authors show the characteristic droplets formed on few-layer BP. Fig 1b and d however show some droplets of water on BP with rounded shape. The authors should discuss this evidence and should report, in the SI, many more BP flakes with droplets on top, in order to have statistical data of elliptic shapes. The OM of the droplets on graphene and MoS2 should be reported in the main text to provide a direct comparison with BP.

Additionally, droplet shapes have been studied only on few-layered materials, (with no specification of the number of layers) while on monolayers the phenomenon might be different as the effect of the substrate may nullify the anisotropic effect, and rounded droplets might be found on BP as well. It would be very important to study droplet formation in monolayered materials since this technique is supposed to be used for lattice orientation identification, and the interesting materials for devices are the atomically thin ones and not the bulk. AFM should be used to characterize the layer numbers of every flake studied. Overall the work discloses an interesting phenomenon from a fundamental point of view, nevertheless there is not sufficient level of novelty and impact to justify publication in Nature Communications. A more specialized journal of high impact would be a more appropriate forum provided that the authors extend the study to monolayer materials and address the points above.

Revisions and list of changes (Manuscript# NCOMMS-19-09038)

Replies to the 1st reviewer's comments (Manuscript# NCOMMS-19-09038)

Comment 1: Figure 1 shows some elliptical shapes of water on black phosphorus. I suggest adding more detailed analysis on the shape evolution and size effect, such as the quantitative data on the shape evolution during the growth process of a water droplet. Even though they are always elliptic during growth, the difference between long axes and short axes of elliptical shape should be different, which depends on the size of water droplet.

Reply 1: Thank you very much for your good suggestion. According to your suggestion, the quantitative statistics on the shape evolution and the corresponding ratio of long axis and short axis for the elliptical water droplets on the surface of BP layer was collected and analyzed. The figure was added to the revised supporting information and the corresponding description was added to the revised manuscript.

Changes in the revised manuscript: A sentence in the second paragraph in page 6, line 114 in original manuscript was changed from “Importantly, water droplets are always elliptic under the whole condensation progress, and all long axes of the water droplets point in almost the same direction, as shown in Fig. 1” to “Statistics, as shown in Fig. S2, have been conducted at 12 s, 13 s, 14 s, 15 s in video S1 to demonstrate the shape evolution of water droplets during the growth process. It demonstrates that the water droplets on the surface of BP layer are either elliptical or spherical, but ellipsoids are dominant. In addition, as is shown in Fig. S2e, the ratios of long axis and short axis for the elliptical water droplets are ranging from 1.2 to 3.0, mainly 1.8 as time goes on. Importantly, all long axes of the water droplets point in almost the same direction, as shown in Fig. 1b and 1d.”

Changes in the revised supporting information: A figure showing data statistics of the shape evolution during the growth process of water droplets was added to the revised supporting information, referring to Figure S2.

Figure S2. The screenshots of different time in video S1, (a) 12 s, (b) 13 s, (c) 14 s, (d) 15 s. (e) The data statistics of the ratio of long axis and short axis for the elliptical water droplets on the BP layer surface at 12 s, 13 s, 14 s, 15 s in video S1. (f) The number of elliptical and spherical water droplets at 12 s, 13 s, 14 s, 15 s in video S1.

Comment 2: Figure 4 indicates that water molecules prefer to align along the zigzag direction than armchair direction since there is an energy difference of 0.1-0.3 eV/molecule between them. It is more convincing to perform some simulations of water on graphene and compare the results of graphene with phosphorene.

Reply 2: Thank you very much for your good suggestion. According to your suggestion, the simulations results of water on graphene and MoS₂ have been added to the supporting information. The corresponding discussion has been added in the revised manuscript.

Changes in the revised manuscript: Sentences have been added to the end of first paragraph in page 9 from line 181-187, which was “Moreover, the same simulations of H₂O molecules on graphene and MoS₂ have been carried out as well. The results demonstrate that both graphene and MoS₂ possess a ground state among the six configurations for H₂O adsorption, with an energy difference of 0.008 - 0.063 eV/molecule for graphene and 0.003 - 0.051 eV/molecule for MoS₂, as shown in Fig. S4 and Fig. S5. However, comparing with BP, the energy difference of graphene and MoS₂ are much smaller, which leads to inconspicuous anisotropy and is in consistent with the experiment results before.”

Changes in the revised supporting information: The simulations of water absorption on graphene and MoS₂ were added in the supporting information, referring to **Figure S4** and **Figure S5**.

Figure S4. Six representative high symmetry configurations are considered for H₂O molecules on graphene, namely, along the armchair (a, b), zigzag (c, d) and diagonal directions (e, f). The energy difference is (a) 0.029 eV/molecule, (b) 0.011 eV/molecule, (c) 0.063 eV/molecule, (d) 0.034 eV/molecule, (e) 0.008 eV/molecule, (f) 0 eV/molecule, respectively.

Figure S5. Six representative high symmetry configurations are considered for H₂O molecules on MoS₂, namely, along the armchair (a, b), zigzag (c, d) and diagonal directions (e, f). The energy difference is (a) 0.011 eV/molecule, (b) 0 eV/molecule, (c) 0.051 eV/molecule, (d) 0.003 eV/molecule, (e) 0.008 eV/molecule, (f) 0.037 eV/molecule, respectively.

Comment 3: Line 90 page 5 mentioned that the anisotropic wetting of water on phosphorene has already been theoretically predicted. However, no citation on the theoretical study of anisotropic wetting of phosphorene is given. It is better to cite some simulation studies on the anisotropic wetting of phosphorene (J. Phys. Chem. C 2018, 122, 4622–4627; Sci. Rep. 2016, 6, 38327) and compare your experimental observations with them, such as the similarities and differences on the shape and energy of water.

Reply 3: Thank you very much for your good suggestion. The referred paper was carefully studied, in which theoretically predicted the anisotropic wetting of water on phosphorene by

using molecular dynamics (MD) simulations. Based on the referred paper, the similarities and differences on the shape and energy of water comparison has been done between my experimental observations and the references, and some discussions according to the references have been added in the revised manuscript.

Changes in the revised manuscript: One sentence was added to the last paragraph in page 8 in line 171, which is “, which is in agreement with the previous simulation results [37].” And sentences were added to the middle of the first paragraph in page 9 from line 176-181, which are “And the result is in accordance with the previous calculation results [36,37] proving that the energy barrier for the water droplets diffusion along the armchair direction is larger than that along the zigzag direction for BP layers, which leads to anisotropic water droplets diffusion. And the anisotropic diffusion of water droplets results in anisotropic shape for water droplets, which normally are elliptical on BP layer surface just like the optical microscope images showing in the experiment.”

Two new references were added in the revised manuscript as [36, 37].

[36] Zhang, W. *et al.* Molecular Structure and Dynamics of Water on Pristine and Strained Phosphorene: Wetting and Diffusion at Nanoscale. *Sci Rep.* **6**, 1-9,(2016).

[37] Chen, S. *et al.* Anisotropic Wetting Characteristics of Water Droplets on Phosphorene: Roles of Layer and Defect Engineering. *The Journal of Physical Chemistry C.* **122**, 4622-4627,(2018).

Replies to the 2nd reviewer's comments (Manuscript# NCOMMS-19-09038)

Comment 1: In the experiment, water vapor was generated and transported to the surface of 2D materials. It is well known that oxygen is dissolved in water at ambient conditions. Thus, it is possible that photo-induced oxidation reaction could occur on BP surface in the presence of oxygen and water. The authors need to clarify the influence of oxidation on liquefaction in this process. Meanwhile, AFM images are necessary to confirm it before and after liquefaction. I cannot find this evidence in current form, which weakens the quality of this manuscript.

Reply 1: Thank you very much for your good suggestion. All the wetting experiments were conducted in a glove box with the protection of inert gas (Ar) mentioned in line 188-190, page 9, not at ambient conditions. And the DI water used in the experiment was eliminated the dissolved O₂ before being put into the glove box. Therefore, it will not lead to photo-induced oxidation reaction on BP surface during the liquefaction process. In addition, the AFM images of BP layer before and after ten times wetting experiments have been provided in the supporting information (original Fig.S4a and S4b), as shown in Figure S3a and S3b, which demonstrates the surface of the BP layer staying the same after ten times wetting experiments, and the Ra can reach 0.22 nm (Figure S3c).

Changes in the revised supporting information: A figure showing the topography of BP layer was added to the revised supporting information, referring to Figure S3.

Figure S3. Atomic force microscope images of BP nanosheets, (a) a pristine BP layer; (b) after ten times water wetting experiments; (c) a zoomed AFM image marked in the yellow dash frame in image b; (d) one day exposed in air; (e) two days exposed in air; (f) a zoomed AFM image in the yellow dash frame in image d.

Comment 2: Previously, researchers have found that rounded and irregularly shaped water droplets were formed on BP surface during microdroplet condensation experiment (2D Mater. 1 (2014) 025001; 2D Mater. 2 (2015) 011002). These data are inconsistent with this work. The authors should justify and clarify it.

Reply 2: Thank you very much for your good suggestion. Two referred papers have been carefully studied. In paper 2D Mater. 1 (2014) 025001, Figure 9, it talked about the microdroplet condensation experiment on the BP surface. What shown in Figure 9b in the paper is consistent with our work to some extent. If the Figure 9b in the paper has been amplified, shown as follow in Figure R1b, many elliptical water droplets can be observed on the surface of BP layer and the long axes of the elliptical water droplets are almost pointing at one direction.

Figure R1. a. Figure 9b image in the paper (2D Mater. 1 (2014) 025001). b. The magnified region marked by the red dash line in Figure 9b in the paper.

In paper 2D Mater. 2 (2015) 011002, the authors claimed the bubbles on the surface of BP exposed in air for a period of time is water droplets from water condensation. However, as some later published papers (Chem. Mater. 2016, 28, 22, 8330-8339 and Nano Lett. 2018, 18, 9, 5618-5627) reported that the bubbles on the surface of BP layer exposed in air for a period of time is not only water, but also the aqueous solution of P_xO_y coming from the oxidation of BP. In this case, it is a different situation like what we talked in the paper.

Comment 3: In line 179, the authors claimed that the BP surface had no bubble remained

after 10 times wetting experiments. However, no detailed or zoomed AFM image was provided to confirm it. In addition, there is a new small flake appeared between the two flakes, located at the left of white line. Possibly it comes from a contaminated AFM tip. In this case, the data should be replaced to avoid misunderstanding.

Reply 3: Thank you very much for your good suggestion. A zoomed AFM image was provided in the Figure S3c to demonstrate there is no bubble remained after ten times wetting experiments. And there is indeed a new small flake appeared between the two flakes located at the left of white line, which may come from a contaminated AFM tip. In this case, a new group of AFM images have been provided in Figure S3a and S3b. The figure was added to the revised supporting information and the corresponding description was added to the revised manuscript.

Changes in the revised manuscript: A sentence in the second paragraph in page 9 from line 178-180 in original manuscript was changed from “Moreover, the surfaces of the BP layers without any bubbles remained very smooth after ten times water wetting experiments, as shown in Fig. S4a and b, which proved that there was no oxidation reaction on the BP surface at all.” to “As the AFM images shown in Fig. S2, the surfaces of the BP layers almost remaining the same without any bubbles after ten times wetting experiments. And the zoomed AFM image in the yellow dashed line frame (Fig. S2c) shows that the Ra can reach 0.22 nm, which is another proof that there was no oxidation reaction on the BP surface at all.”

Changes in the revised supporting information: A figure showing the topography of BP layer was added to the revised supporting information, referring to Figure S3.

Figure S3. Atomic force microscope images of BP nanosheets, (a) a pristine BP layer; (b) after ten times water wetting; (c) a zoomed AFM image marked in the yellow dash frame in image b; (d) one day exposed in air; (e) two days exposed in air; (f) a zoomed AFM image in the yellow dash frame in image d.

Comment 4: In figure 5, only Raman data are not enough to demonstrate the nondestructive performance of liquefaction for BP surface. The authors should provide more characterization data to support it, such as AFM image.

Reply 4: Thank you very much for your good suggestion. The AFM images of BP layer before and after ten times wetting experiments have been provided in the supporting information, Figure S3, which demonstrates the nondestructive performance of liquefaction for BP surface. To further demonstrate the nondestructive performance of liquefaction for BP surface, a few-layer BP FET device was fabricated and the transport performance has been tested before and after water liquefaction experiment. It shows that the transport performance, like high mobility and on/off ratio, remains almost the same after ten times wetting experiments. The figure has been added to the revised supporting information and the corresponding description has been added to the revised manuscript.

Changes in the revised manuscript: Two sentences were added to the middle of the paragraph in page 10 from line 206-210, which are “Furthermore, a few-layer BP based FET device (~ 8.34 nm) was fabricated, as shown in the inset in Fig. S6b, the transport performance has been tested before and after water liquefaction experiment. The result shows that the I_d - V_g curves almost remain the same before (black line) and after ten times wetting experiments (red line), which

also demonstrates that there was no structural collapse existing on the BP after ten times wetting experiments.”

Changes in the revised supporting information: A figure showing the topography of BP layer was added to the revised supporting information, referring to Figure S3. And the electrical properties based on a few-layer BP FET before and after ten times wetting experiments was added to supporting information, referring to Figure S6.

Figure S3. Atomic force microscope images of BP nanosheets, (a) a pristine BP layer; (b) after ten times water wetting experiments; (c) a zoomed AFM image marked in the yellow dash frame in image b; (d) one day exposed in air; (e) two days exposed in air; (f) a zoomed AFM image in the yellow dash frame in image d.

Figure S6. The V_g and I_d curves obtained from the BP FET (inset image b) of pristine BP layer (black line) and after ten times wetting experiments BP layer (red line).

Comment 5: In line 190, the authors claimed many bubbles were left on BP surface after exposure to air for 2 days. However, these structures were attributed to water droplets in the literatures (2014) 025001; 2D Mater. 2 (2015) 011002). How can the authors prove they are bubbles?

Reply 5: Thank you very much for your good question. The referred two papers 2D Mater. (2014) 025001 and 2D Mater. 2 (2015) 011002) were fully studied and were cited in our paper. In paper 2D Mater. 1 (2014) 025001, Figure 9, it really said that the water droplets will form on the surface of BP from air moisture condensation in very short time. However, in our case and in paper 2D Mater. 2 (2015) 011002, the BP layers were put in air for 1 and 2 days or longer time. During this period of time exposed in air, degradation will happen on the surface of BP under ambient conditions together with the water condensation. The degradation product of BP exposed in air for period of time is mainly P_xO_y according to the articles (Chem. Mater. 2016, 28, 22, 8330-8339 and Nano Lett. 2018, 18, 9, 5618-5627). And in the paper 2D Mater. 2 (2015) 011002, the authors named the degradation product on the BP layer surface as white bubble in the AFM image. In this case, we said that there are many bubbles on the surface of BP after two days exposure to the air.

Changes in the revised manuscript: Two new references were added in the revised manuscript as [38], [46].

[38] Castellanos-Gomez, A. *et al.* Isolation and characterization of few-layer black phosphorus. *2D Materials*. **1**, 025001,(2014).

[46] Joshua O Island, G. *et al.* Environmental instability of few-layer black phosphorus. *2D Materials*. **2**, 1-6,(2015).

Comment 6: Although the idea is interesting, I think the work has weakness in the characterization for surface investigation. In my opinion, AFM and other instruments should be necessary to monitor the evolution before and after liquefaction to provide solid evidence. Only optical microscope and Raman spectrometer are insufficient. The authors almost reveal the final result but lots of works are remained to be conducted.

Reply 6: Thank you very much for your good suggestion. Your suggestion is very constructive, However, to be honest, it is hard for us to monitor the evolution progress by AFM because the liquefaction progress of water on the BP layer surface is very fast (< 10 s), like Video S1 shows.

We cannot obtain one AFM image in one or two seconds. So, if you have any other better recommended instruments to monitor the liquefaction progress, we are willing to adopt it to improve the quality of our paper.

Comment 7: Correct the format of reference.

Reply 7: Thank you very much for your good suggestion. According to your suggestion, the format of the reference has been carefully corrected in the revised manuscript.

Replies to the 3rd reviewer's comments (Manuscript# NCOMMS-19-09038)

Comment 1: In Fig 1 the authors show the characteristic droplets formed on few-layer BP. Fig 1b and d however show some droplets of water on BP with rounded shape. The authors should discuss this evidence and should report, in the SI, many more BP flakes with droplets on top, in order to have statistical data of elliptic shapes. The OM of the droplets on graphene and MoS₂ should be reported in the main text to provide a direct comparison with BP.

Reply 1: Thank you very much for your good suggestion. According to your suggestion, the quantitative statistics on the shape evolution and the corresponding ratio of long axis and short axis for the elliptical water droplets on the BP layer surface was collected and analyzed. The figure was added to the revised supporting information and the corresponding description was added to the revised manuscript. In addition, the OM images of the droplets on graphene and MoS₂ have been changed to the main text to provide a direct comparison with BP, referring to Figure 1.

Changes in the revised manuscript: A sentence in the second paragraph in page 6, line 114 in original manuscript was changed from “Importantly, water droplets are always elliptic under the whole condensation progress, and all long axes of the water droplets point in almost the same direction, as shown in Fig. 1” to “Statistics, as shown in Fig. S2, have been conducted at 12 s, 13 s, 14 s, 15 s in video S1 to demonstrate the shape evolution of water droplets during the growth process. It demonstrates that the water droplets on the surface of BP layer are either elliptical or spherical, but ellipsoids are dominant. In addition, as is shown in Fig. S2e, the ratios of long axis and short axis for the elliptical water droplets are ranging from 1.2 to 3.0, mainly 1.8 as time goes on. Importantly, all long axes of the water droplets point in almost the same direction, as shown in Fig. 1b and 1d.” The former OM images of the droplets on graphene and MoS₂ in the supporting information named **Figure S2** and **Figure S3** have been moved to the main text referring to **Fig. 1** to provide a direct comparison with BP.

Fig. 1 Optical images of few-layer BP/graphene/MoS₂ without(a/e/i) and with (b/f/j) water droplets on their surface. Optical images of thick BP /graphite/MoS₂ without (c/g/k) and with (d/h/l) water droplets on their surface.

Changes in the revised supporting information: A figure showing data statistics of the shape evolution during the growth process of water droplets was added to the revised supporting information, referring to Figure S2.

Figure S2. The screenshots of different time in video S1, (a) 12 s, (b) 13 s, (c) 14 s, (d) 15 s. (e) The data statistics of the ratio of long axis and short axis for the elliptical water droplets on the BP layer surface at 12 s, 13 s, 14 s, 15 s in video S1. (f) The number of elliptical and spherical water droplets at 12 s, 13 s, 14 s, 15 s in video S1.

The former OM images of the droplets on graphene and MoS₂ in the supporting information named **Figure S2** and **Figure S3** have been deleted.

Comment 2: Additionally, droplet shapes have been studied only on few-layered materials, (with no specification of the number of layers) while on monolayers the phenomenon might be different as the effect of the substrate may nullify the anisotropic effect, and rounded droplets might be found on BP as well. It would be very important to study droplet formation in monolayered materials since this technique is supposed to be used for lattice orientation identification, and the interesting materials for devices are the atomically thin ones and not the bulk. AFM should be used to characterize the layer numbers of every flake studied.

Reply 2: Thank you very much for your good suggestion. I fully agree with what you mentioned that there may be more rounded droplets to be found on monolayer BP due to the effect of the substrate. The reason why the few-layered materials studied in the paper are in 8~10 nm thickness is that 2D layered materials in such thickness have been most extensively studied and possess relatively better performance for electronic and optoelectronic devices research than monolayer ones^[1,2], especially for BP layers^[3-5]. And the AFM have been used to characterize the layer numbers of the flake studied in the paper.

Changes in the revised manuscript: A sentence was added to the end of first paragraph in page 12 from line 240-243, which is “Additionally, all the few-layered materials studied here are in 8~10 nm thickness, because 2D layers materials in such thickness have been most extensively studied and possess relatively better performance for devices research than monolayer ones, especially for BP layers^[47-50].”

Changes in the revised supporting information: The thickness information of the BP flake has been added to Figure S3.

Figure S3. Atomic force microscope images of BP nanosheets, (a) a pristine BP layer; (b) after ten times water wetting experiments; (c) a zoomed AFM image marked in the yellow dash frame in image b; (d) one day exposed in air; (e) two days exposed in air; (f) a zoomed AFM image in the yellow dash frame in image d.

Comment 3: Overall the work discloses an interesting phenomenon from a fundamental point of view, nevertheless there is not sufficient level of novelty and impact to justify publication in Nature Communications. A more specialized journal of high impact would be a more appropriate forum provided that the authors extend the study to monolayer materials and address the points.

Reply 3: Thank you very much for your comments. There were three reviewers who reviewed this submission to Nature Communications. Another two reviewers also thought that this work is very interesting as you, and would be suitable for publishing on Nature Communications with some modifications. In this case, the manuscript has been carefully revised according to all the comments provided by the reviewers. So, I really hope you can reconsider our work from the revised manuscript. Thanks again.

References

- ¹ Yuan, H. *et al.* Polarization-sensitive broadband photodetector using a black phosphorus vertical p–n junction. *Nature Nanotechnology*. **10**, 707-714,(2015).
- ² Guo, Q. *et al.* Black Phosphorus Mid-Infrared Photodetectors with High Gain. *Nano Lett.* **16**, 4648-4655,(2016).
- ³ Li, L. *et al.* Black phosphorus field-effect transistors. *Nat Nanotechnol.* **9**, 372-377,(2014).
- ⁴ Liu, H. *et al.* Phosphorene: an unexplored 2D semiconductor with a high hole mobility. *ACS Nano*. **8**, 4033-4041,(2014).
- ⁵ Liu, H. *et al.* Semiconducting black phosphorus: synthesis, transport properties and electronic applications. *Chem Soc Rev.* **44**, 2732-2743,(2015).

REVIEWERS' COMMENTS:

Reviewer #1 (Remarks to the Author):

The authors have improved the manuscript significantly. It is recommended for publication on Nature Communications.

Reviewer #2 (Remarks to the Author):

The authors have well addressed my concerns. I recommend acceptance now.

Reviewer #3 (Remarks to the Author):

The authors have now addressed the point raised by the reviewers and the manuscript is much stronger in supporting the thesis of elliptic droplets and now recommendable for publication.

I still think that showing the evolution of the droplet shape from mono- to few- layers up to bulk would have been a useful addition to the Manuscript which would have linked this work to previous work on mono-few-layered graphene.

Point-by-point response to reviewers (Manuscript# NCOMMS-19-09038A)

Reviewer #1 (Remarks to the Author):

The authors have improved the manuscript significantly. It is recommended for publication on Nature Communications.

No comment.

Reviewer #2 (Remarks to the Author):

The authors have well addressed my concerns. I recommend acceptance now.

No comment.

Reviewer #3 (Remarks to the Author):

The authors have now addressed the point raised by the reviewers and the manuscript is much stronger in supporting the thesis of elliptic droplets and now recommendable for publication.

Replies to the 3st reviewer's comment.

Comment 1: I still think that showing the evolution of the droplet shape from mono- to few- layers up to bulk would have been a useful addition to the Manuscript which would have linked this work to previous work on mono-few-layered graphene.

Reply 1: Thank you very much for your good suggestion. The reason why the few-layered materials studied in the paper are in 8~10 nm thickness is that 2D layered materials in such thickness have been most extensively studied and possess relatively better performance for electronic and optoelectronic devices research than monolayer ones, especially for BP layers. And to be honest, it is very difficult to obtain a big enough mono-layered BP through a scotch tape-based mechanical exfoliation method to observe the water liquefaction progress by optical microscope now. However, we will try our best to do such experiment in our future work. Thanks again for your good suggestion and agreement to accept our paper.